# Adrenal Incidentaloma: From Silent Diagnosis to Clinical Challenge

**DOI:** 10.3390/biomedicines13092298

**Published:** 2025-09-19

**Authors:** Alexandra Mirica, Dana-Mihaela Tilici, Diana Loreta Paun, Ana Maria Arnautu, Victor Nimigean, Sorin Paun

**Affiliations:** 1Endocrinologie, CHU Brugmann, 1020 Bruxelles, Belgium; dr.alexandramirica@gmail.com; 2Doctoral School of “Carol Davila” University of Medicine and Pharmacy, 020021 Bucharest, Romania; 3Endocrinology Department, Bucharest University Emergency Hospital, Splaiul Independenței 169, 050098 Bucharest, Romania; diana.paun@umfcd.ro (D.L.P.);; 4Department of Anatomy, Faculty of Dentistry, Carol Davila University of Medicine and Pharmacy, 050474 Bucharest, Romania; victor.nimigean@umfcd.ro; 5Bucharest Emergency Clinical Hospital, Calea Floreasca 8, 014461 Bucharest, Romania; sorin.paun@umfcd.ro

**Keywords:** adrenal incidentaloma, autonomous cortisol secretion, adrenal adenoma, endocrine imaging, adrenalectomy, subclinical Cushing’s syndrome

## Abstract

The widespread use of advanced imaging techniques has led to a rising incidence of adrenal incidentalomas (AIs), asymptomatic adrenal masses discovered during imaging for non-adrenal-related conditions. AIs represent a diagnostic and therapeutic challenge due to their varied etiology, secretory potential, and potential for malignancy. This review aims to provide a comprehensive overview of the current knowledge on adrenal incidentalomas, focusing on their pathogenesis, diagnostic work-up, imaging features, hormonal evaluation, and evidence-based management, with a special emphasis on autonomous cortisol secretion (ACS). A thorough narrative review of the literature from the past two decades was conducted, synthesizing data from key international guidelines (ESE/ENSAT), observational studies, meta-analyses, and case series regarding the evaluation and treatment of AI. AI represents an increasingly relevant clinical condition requiring a multidisciplinary, personalized approach. Prompt endocrine and radiological evaluation is essential to identify hormonally active or potentially malignant tumors. The complexity of the natural history of AI and the evolving understanding of ACS underline the need for tailored follow-up and management strategies.

## 1. Introduction

Over the last four decades, the dramatic evolution of cross-sectional imaging has markedly increased visualization of the adrenal glands and, consequently, the incidental detection of adrenal masses—a phenomenon that has transformed adrenal disease from a relatively rare diagnostic curiosity into a common clinical problem.

An adrenal incidentaloma (AI) is conventionally defined as an asymptomatic adrenal lesion ≥ 1 cm discovered unexpectedly on imaging performed for indications unrelated to suspected adrenal disease; this definition expressly excludes lesions detected through targeted screening of patients with genetic syndromes that predispose to adrenal tumors and lesions identified during staging or surveillance of known extra-adrenal malignancy [1].

Although most AIs represent benign, non-functioning adrenocortical adenomas, a clinically meaningful minority may be hormonally active or harbor malignant potential (including metastases, pheochromocytoma or adrenocortical carcinoma). Discriminating these entities requires synthesis of high-quality morphological imaging and directed biochemical testing, interpreted in the light of clinical context, comorbidity, and lesion dynamics. Per the recent European Society of Endocrinology guidelines, this integrative strategy—rather than any single test—best balances the twin goals of avoiding unnecessary interventions while not missing clinically significant disease [2].

Because diagnostic uncertainty persists for a subset of lesions, contemporary care pathways favor structured, multidisciplinary evaluation (radiology, endocrinology, surgery, and pathology), with individualized decisions on surveillance intervals, further imaging or functional studies, and thresholds for referral to surgery. Emerging modalities and quantitative tools (advanced imaging metrics, radiomics, and steroid metabolomics) are beginning to complement conventional approaches and may refine risk stratification in the near future, although their routine clinical role remains under evaluation [2].

Finally, a focused discussion will follow addressing the comparative questions and remaining uncertainties around minimally invasive partial versus total adrenalectomy for unilateral primary hyperaldosteronism, including issues of outcome equivalence, case selection bias toward smaller tumors, and heterogeneity in reporting [3].

## 2. Materials and Methods

### 2.1. Study Design and Search Strategy

We conducted a narrative literature review to examine the epidemiology, pathogenesis, imaging, biochemical evaluation, and management of AIs. A comprehensive search was performed in PubMed/MEDLINE, Embase, Scopus, Web of Science, and the Cochrane Library up to 1 August 2025. Search terms included both controlled vocabulary and free-text terms related to “adrenal incidentaloma” and associated functional entities (e.g., autonomous cortisol secretion, pheochromocytoma, primary aldosteronism, adrenocortical carcinoma), combined with imaging and diagnostic keywords. Additionally, reference lists of included studies and relevant guidelines were manually screened to identify further pertinent sources.

### 2.2. Eligibility Criteria and Study Selection

We included clinical studies, guidelines, systematic reviews, and case reports pertinent to AI in human populations. Exclusion criteria encompassed animal or in vitro studies lacking clinical relevance, conference abstracts without full-text availability, and reports deemed non-relevant. This narrative review was conducted based on a comprehensive PubMed search performed by two independent authors.

### 2.3. Data Extraction and Synthesis

Key information extracted from the selected studies encompassed study design, population characteristics, diagnostic definitions, imaging and biochemical criteria, management strategies, and reported outcomes. Data were synthesized qualitatively, with findings compared against current clinical guidelines. No meta-analysis was conducted.

## 3. Results

### 3.1. Prevalence

Adrenal masses are among the most frequently encountered lesions of the adrenal glands, and their detection has surged dramatically over the past two decades, reflecting the widespread adoption of high-resolution imaging technologies in routine clinical practice [2]. Contemporary studies indicate that approximately three-quarters of adrenal incidentalomas are discovered unexpectedly during imaging performed for unrelated indications. With the use of modern CT and MRI scanners, prevalence estimates in adults range from 4 to 5%, rising to 7–10% in older populations. While the majority of incidentalomas are unilateral, bilateral lesions occur in 10–15% of cases [1,2,4,5]. The incidence peaks in the fifth and sixth decades, whereas lesions in individuals under 30 and pediatric patients remain uncommon [6]. This trend underscores the evolving significance of adrenal incidentalomas as a pressing public health consideration and a common clinical challenge [1,7].

Management of adrenal incidentalomas is nuanced, reflecting the dual need to evaluate malignant potential and endocrine activity. Although most lesions are nonfunctioning adrenocortical adenomas, a notable subset can harbor significant hormonal activity or malignancy, including pheochromocytomas, adrenocortical carcinomas, hormone-secreting adenomas, or metastatic deposits [2]. Rarely, apparently benign lesions may transform into metastatic disease, emphasizing that even seemingly indolent adrenal incidentalomas can follow unpredictable trajectories [8].

Clinical assessment of an incidentaloma is guided by two central questions: does the lesion demonstrate radiologic or clinical features suspicious for malignancy, and does it exhibit secretory activity with potential systemic consequences? Current international guidelines advocate a structured, evidence-based approach to these questions, integrating imaging characteristics, hormonal evaluation, and patient-specific factors to inform individualized management strategies [2]. This approach not only directs immediate therapeutic decisions but also shapes long-term surveillance, balancing patient safety, procedural risks, and the growing need for precision in adrenal disease management.

### 3.2. Pathogenesis

Multiple mechanisms have been described in the pathogenesis of AI, but the molecular mechanisms that promote AI development are not well understood. Firstly, one of the pathophysiological mechanisms described in the literature is focal adrenal hyperplasia in response to ischemic lesions, as a natural phenomenon of aging. This mechanism could be a rational explanation for the increasing prevalence of AI among the elderly [9,10].

Secondly, increased insulin levels can induce AI development through mitogenic action on the adrenal cortex. This mechanism is cited in cases of AI in overweight people with insulin resistance [10].

Thirdly, changes in the sensitivity of the hypothalamic–pituitary–adrenal axis, which may lead to subtle but chronic stimulation of the adrenals by constantly elevated levels of ACTH, particularly in response to chronic stress, can lead to the development of nodular adrenal hyperplasia [11]. Recent studies have elucidated several molecular mechanisms contributing to AI development. In primary bilateral macronodular adrenal hyperplasia (PBMAH), a rare cause of primary overt Cushing’s syndrome, up to one-third of bilateral AIs with evidence of cortisol excess exhibit increased steroidogenesis regulated by aberrantly expressed G protein-coupled receptors (GPCRs). Some receptor ligands are ectopically produced within PBMAH tissues, creating autocrine/paracrine regulation of steroidogenesis. Additionally, germline heterozygous inactivating mutations of the ARMC5 gene have been identified in 20% to 25% of apparent sporadic cases and more frequently in familial cases. These mutations are associated with PBMAH and can also be linked to meningiomas. More recently, combined germline mutations and somatic events inactivating the KDM1A gene have been specifically identified in patients with glucose-dependent insulinotropic peptide (GIP)-dependent PBMAH. Functional studies demonstrated that inactivation of KDM1A leads to GIP-receptor overexpression and dysregulation of other GPCRs [11].

Furthermore, somatic mutations in other genes, such as KCNJ5, CACNA1D, and ATP1A1, have been implicated in aldosterone-producing adenomas, highlighting the complex genetic landscape of adrenal tumors. These molecular insights underscore the importance of a comprehensive approach to evaluating AIs, integrating clinical, biochemical, radiological, and genetic data to guide management decisions effectively [11,12,13].

In addition to somatic mutations, several hereditary syndromes are associated with adrenal tumor development, further highlighting the complex genetic landscape of adrenal incidentalomas. McCune-Albright syndrome results from somatic activating mutations in the GNAS gene and is characterized by polyostotic fibrous dysplasia, café-au-lait spots, and endocrine hyperfunction; approximately 5% of affected patients develop cortisol-secreting adrenal lesions, often presenting as macronodular adrenal hyperplasia [9,11]. Beckwith-Wiedemann syndrome, caused by genetic and epigenetic alterations at 11p15.5, is associated with overgrowth and an increased risk of adrenocortical carcinoma and other embryonal tumors, although adrenal tumors remain rare [11,14].

Li-Fraumeni syndrome, due to germline TP53 mutations, confers a high risk of multiple malignancies, including adrenocortical carcinoma, particularly in children and young adults [11,13].

MEN1, caused by mutations in the MEN1 gene, classically leads to parathyroid, pituitary, and pancreatic neuroendocrine tumors, but adrenal tumors—predominantly non-functioning—occur in up to 40% of patients [15,16,17,18]. MEN4, associated with CDKN1B mutations, similarly predisposes to endocrine and non-endocrine tumors, including adrenal adenomas and rare cortisol-secreting lesions [19,20]. Moreover, inactivating mutations in PRKAR1A, initially described in Carney complex, can result in constitutive activation of the cAMP–protein kinase A pathway, leading to cortisol-producing adrenal tumors in sporadic cases [6].

Beyond genetic factors, environmental contributors such as exposure to endocrine-disrupting chemicals, notably bisphenol A (BPA), have been implicated in AI development, although current evidence remains preliminary. Collectively, these findings underscore that adrenal 76 alomas arise from a multifactorial interplay of somatic mutations, hereditary syndromes, and environmental influences, reinforcing the need for a comprehensive evaluation that integrates clinical, biochemical, radiological, and genetic data to inform personalized management strategies [21,22].

### 3.3. Diagnostic Features

Along with ruling out neoplasia, determining the endocrine state of individuals with AI is a critical component of clinical management. Personalized care is supported by a hormonal evaluation, clinical examination correlated with possible excess hormone secretion, focusing on suggestive signs and symptoms and related comorbidities [2,6].

According to a study regarding AI, adenomas account for 41% of total AI, metastases for 19%, adrenal cortical carcinoma for 10%, myelolipomas for 9%, and pheochromocytomas for 8%, with other, more rare entities such as adrenal cysts, ganglioneuromas, hematomas, and infectious or infiltrative lesions. With a prevalence ranging from 5% to 30%, cortisol excess is the most prevalent endocrine abnormality in people with AI [10,23]. In adults, an AI can represent many endocrine disorders, which are synthesized in Table 1 (adapted from [5]).

Alongside excluding malignancy, assessing the endocrine function of patients with AIs is a central aspect of clinical management [2,6]. Current international guidelines recommend a structured hormonal evaluation for all patients, including:-Cortisol excess: overnight 1 mg dexamethasone suppression test (DST), with a post-DST serum cortisol > 1.8 µg/dL (50 nmol/L) suggesting autonomous cortisol secretion; additional testing may include late-night salivary cortisol or 24 h urinary free cortisol when indicated.-Primary hyperaldosteronism: in patients with hypertension or hypokalemia, measurement of plasma aldosterone concentration (PAC) and plasma renin activity (PRA) to calculate the aldosterone-to-renin ratio (ARR), with confirmatory testing per local protocols.-Pheochromocytoma: plasma-free metanephrines or 24 h urinary fractionated metanephrines in all patients with adrenal masses, especially when radiological features are suggestive or symptoms such as paroxysmal hypertension, palpitations, or headaches are present.

Epidemiological studies indicate that adenomas account for approximately 40–45% of AIs, metastases for 15–20%, adrenocortical carcinoma for 5–10%, myelolipomas for 5–10%, and pheochromocytomas for 5–10%, with rarer lesions including cysts, ganglioneuromas, hematomas, and infectious or infiltrative processes. Among endocrine abnormalities, autonomous cortisol secretion is the most common, reported in 5–30% of patients depending on population characteristics [5,10,23]. A summary of potential endocrine disorders associated with AI, based on recent literature and international guideline recommendations, is provided in Table 1.

In addition, one case of AI was reported, which on pathological examination proved to be Kaposi’s sarcoma [24]. Similarly, adrenal oncocytoma may be a rare etiology of AI [25]. The endocrine assessment should exclude pheochromocytomas, cortisol-producing adenomas, and steroid-producing tumors.

Cortisol excess is the most frequent endocrine abnormality among patients with AIs. According to the 2023 international guidelines, all patients with AI should undergo an overnight 1 mg dexamethasone suppression test (DST) to screen for cortisol hypersecretion [2]. Post-DST serum cortisol ≤ 1.8 µg/dL (50 nmol/L) generally excludes autonomous cortisol secretion (ACS).

The concept of autonomous cortisol secretion (ACS) has replaced the older term “subclinical Cushing’s syndrome.” ACS refers to patients with biochemical evidence of cortisol excess without classical Cushingoid features, such as central obesity, moon face, myopathy, and skin fragility. Patients with post-DST cortisol levels between 1.9 and 5 µg/dL (105–138 nmol/L) are classified as having possible autonomous cortisol secretion (PACS). Confirmatory testing for overt cortisol excess may include late-night salivary cortisol, 24 h urinary free cortisol (UFC), or a 2-day low-dose dexamethasone suppression test, with positivity in at least two of three tests supporting the diagnosis of overt Cushing’s syndrome [2,26].

In ACTH-independent cortisol excess due to adrenal tumors, typical biochemical features include suppressed plasma ACTH, reduced DHEAS, elevated UFC, and lack of cortisol suppression on DST. Interpretation may be complicated by physiological hypercortisolism related to stress, obesity, or comorbidities, and by assay variability among immunoassays, chemiluminescence, or mass spectrometry methods [27,28].

ACS and PACS are clinically significant because even mild cortisol excess is associated with increased cardiovascular risk, metabolic disorders (type 2 diabetes, dyslipidemia, obesity), subclinical atherosclerosis, thromboembolic events, sarcopenia, and vertebral fractures. Patients with ACS/PACS should therefore undergo comprehensive metabolic and cardiovascular assessment, with ongoing monitoring for blood pressure, glycemic control, lipid profile, and bone health [2,27,28].

Progression from ACS or PACS to overt Cushing’s syndrome is very rare (<0.1%), but patients may still experience substantial morbidity and increased mortality [29,30,31,32]. Overall, up to 30% of patients with AI may exhibit ACS, reinforcing the guideline recommendation for routine dexamethasone testing in all adrenal incidentalomas and highlighting the importance of integrating clinical, biochemical, and radiological data for risk stratification and personalized management [6,33].

Early identification of pheochromocytoma is critical, as undiagnosed or untreated cases are associated with significant cardiovascular morbidity and mortality [34]. Current guidelines recommend screening all patients with AIs for pheochromocytoma using plasma-free metanephrines or 24 h urinary fractionated metanephrines. While some authors suggest that screening may be unnecessary in patients with lipid-rich cortical adenomas on CT, additional imaging features such as tumor heterogeneity, necrosis, or advanced age should prompt testing regardless of attenuation values [2,6].

Care must be taken to minimize false-positive results, as several medications (e.g., sympathomimetics, glucocorticoids, tricyclic antidepressants, antipsychotics, acetaminophen) and certain foods (bananas, cheese, coffee, cured meats, chocolate, soy sauce, red wine, fava beans, and yogurt) can interfere with testing [35,36]. Plasma samples are ideally collected in the supine position after a period of rest, and concurrent measurement of urinary creatinine is recommended. In suspected dopamine-secreting tumors, measurement of plasma 3-methoxytyramine is advised [37,38,39].

If biochemical testing shows metanephrine levels threefold or higher than the upper reference limit, localization imaging should follow [40]. Contrast-enhanced CT is typically the first-line modality, with MRI or ^123^I-metaiodobenzylguanidine (MIBG) scintigraphy recommended for further characterization or detection of metastatic disease [41]. Clinical vigilance is important, as asymptomatic pheochromocytomas have been reported even in young patients, highlighting the need for strict adherence to hormonal evaluation protocols [42].

Primary hyperaldosteronism (PA) was believed to be relatively uncommon in the context of AIs. Screening for PA is critically underperformed and often delayed, frequently occurring years after the initial diagnosis of hypertension, typically after severe complications have emerged. This delay is partly due to the misconception that PA is only present in patients with hypokalemia, adrenal macronodules, markedly elevated aldosterone levels, or severe hypertension. Consequently, many patients are treated empirically for primary hypertension, missing opportunities for targeted therapies or potential cures, while continuing to experience suboptimally controlled blood pressure and increased cardiovascular and renal risk. The 2024 European Society of Cardiology (ESC) guidelines and 2025 Primary Aldosteronism Endocrine Society Clinical Practice Guideline recommendations emphasize screening for PA in all adults with diagnosed hypertension, given that PA is a common cause of secondary hypertension and is associated with higher cardiovascular risk compared with primary hypertension [43,44,45,46]. Appropriate, guideline-directed management—including unilateral adrenalectomy for lateralizing disease or mineralocorticoid receptor antagonists (MRAs) for bilateral disease—can effectively normalize blood pressure, correct hypokalemia, and reduce excess cardiovascular morbidity, yet these therapies remain underutilized due to insufficient routine screening [45,46].

In patients with hypertension or unexplained hypokalemia, the ESE/ENSAT guidelines recommend screening using the aldosterone-to-renin ratio (ARR) under standardized conditions, followed by confirmatory testing when indicated. Initial evaluation may include plasma aldosterone concentration, plasma renin activity, or direct renin concentration, with ARR sensitivity ranging from 68 to 94% and a negative predictive value approaching 100%. Before testing, serum potassium should be normalized, and interfering medications—including ACE inhibitors, ARBs, beta-blockers, and diuretics—should be withdrawn according to recommended intervals (2–6 weeks) [2,29,47].

Elevated sex steroids or steroid precursors in patients with suspicious clinical or imaging features are highly suggestive of adrenocortical carcinoma and warrant thorough diagnostic evaluation [2]. Androgen-secreting tumors in women may present with virilization: hirsutism, acne, deepened voice, clitoromegaly, male-pattern baldness, and primary amenorrhea in adolescents. Estrogen-secreting tumors in women may cause irregular uterine bleeding or mastalgia, whereas men may exhibit feminization, including gynecomastia, testicular regression, and reduced libido.

To avoid unnecessary adrenalectomy, congenital adrenal hyperplasia (CAH) should be considered in patients with AIs, particularly in cases of bilateral adrenal masses, with measurement of 17-hydroxyprogesterone recommended when clinically indicated [46].

Finally, adrenal insufficiency can occur in patients with bilateral adrenal lesions, including large bilateral adrenal metastases. Consistent with recent guidelines, all patients at risk should undergo appropriate adrenal function testing, particularly before surgical intervention or initiation of therapies that may further compromise adrenal reserve [2,27].

Recent advancements in the assessment of adrenal incidentalomas (AIs) have led to revised imaging protocols aimed at reducing unnecessary interventions while ensuring accurate diagnosis. The current standard for initial evaluation involves non-contrast computed tomography (CT). A homogeneous adrenal mass with an attenuation value of ≤10 Hounsfield units (HU) on non-contrast CT is typically indicative of a benign, lipid-rich adenoma, and no further imaging is necessary in such cases [29,47,48].

In contrast, lesions with attenuation values > 10 HU warrant additional evaluation. For masses larger than 4 cm or those exhibiting inhomogeneity, a contrast-enhanced CT scan with washout analysis is recommended. A relative washout of >40% or an absolute washout of >60% in the delayed phase suggests a benign lesion, with sensitivity and specificity exceeding 90% [29,34,49]. However, recent studies have questioned the necessity of washout CT in the evaluation of AIs. A study published in 2025 proposed eliminating washout CT and increasing the CT attenuation threshold to 20 HU for lesions < 4 cm, arguing that this approach effectively excludes malignancy in the majority of cases [47,48].

In cases where imaging features are indeterminate or suggest malignancy, further evaluation with positron emission tomography (PET) or PET/CT may be considered [2,50]. Fine needle aspiration (FNA) biopsy is generally reserved for specific circumstances, such as when metastasis is suspected, and should only be performed after biochemical exclusion of pheochromocytoma [51].

Regarding the risk of malignancy, studies have shown that the likelihood of adrenocortical carcinoma (ACC) increases with tumor size. For lesions < 4 cm, the risk is approximately 2%, rising to 6% for those between 4.1 and 6 cm, and 25% for lesions > 6 cm [33,48,52,53]. Additionally, lesions with attenuation values > 20 HU on non-contrast CT are associated with a higher risk of malignancy [48,54].

In summary, the updated guidelines advocate for a streamlined approach to the evaluation of AIs, emphasizing the importance of non-contrast CT with appropriate attenuation thresholds and selective use of advanced imaging techniques to minimize unnecessary procedures while effectively identifying potential malignancies.

### 3.4. Management and Follow-Up of AI

The management of adrenal incidentalomas (AIs) is tailored to tumor type, patient comorbidities, and individual preferences. Surgical intervention remains the primary treatment for specific adrenal lesions, but it requires careful patient selection and should ideally be performed at high-volume centers by surgeons experienced in both laparoscopic and open adrenal procedures, with a minimum of 15 adrenalectomies per year to ensure optimal outcomes and minimize complications [55,56].

Current evidence supports surgery for hyperfunctioning adrenal tumors, masses suspicious for malignancy, and indeterminate lesions. Laparoscopic adrenalectomy is generally recommended for unilateral lesions with benign imaging characteristics or for tumors ≤ 6 cm that appear suspicious on imaging but show no evidence of local invasion. Open adrenalectomy is reserved for lesions with imaging features suggestive of malignancy accompanied by evidence of local tissue invasion. For patients with ACS who have associated comorbidities potentially related to cortisol excess, individualized surgical consideration is advised. In these cases, ongoing risk assessment and annual clinical follow-up for up to five years are recommended to guide decision-making regarding possible surgery [2,3,50,57,58,59,60].

Guidelines also emphasize avoiding surgery in asymptomatic, non-functioning, unilateral AIs with clearly benign imaging features. In bilateral AIs, subclinical hypercortisolism appears more frequently than in unilateral lesions, but whether surgery is necessary for all cases or which adrenal should be resected first remains uncertain. Preoperative management for patients with ACS or possible ACS should include perioperative glucocorticoid coverage during major surgical stress [6,57,60,61].

When a decision is made to monitor an AI, follow-up imaging is typically performed every 6–12 months, adjusted based on tumor characteristics. Surgery should be considered if the lesion enlarges by more than 20% in volume and at least 5 mm in maximum diameter, or if signs of hormonal excess emerge. If growth remains below these thresholds, repeat imaging after 6–12 months is recommended. Hormonal reevaluation is generally deferred unless new clinical indicators or worsening comorbidities arise. The same principles apply to bilateral AIs, with comorbidity assessment and endocrine evaluation following a similar framework [2,59].

Special consideration is warranted for high-risk populations, including young adults under 40, children, and pregnant women, who require prompt evaluation, preferably with MRI over CT to minimize radiation exposure. Meta-analyses of AI natural history indicate that non-functioning AIs and those with mild autonomous cortisol excess rarely exhibit significant changes in size or function over time. The probability of progressing to clinically overt Cushing’s syndrome is very low, estimated at approximately 0.1% [30].

Overall, the contemporary management strategy emphasizes individualized care, guided by tumor biology, patient comorbidities, and risk stratification, with multidisciplinary input to optimize both clinical outcomes and patient safety.

### 3.5. Clinical Implications of AI

The medical literature abounds with numerous studies conducted to evaluate the hypothesis that non-functioning AI increases the risk of cardiovascular events and has significant metabolic consequences compared to the absence of adrenal tumors. In that direction, results from a study published in 2019 indicate that vascular changes precede the development of cardiovascular disease and may increase morbidity and mortality in patients with AI. In addition, the same authors conclude that apparently hormonally inactive adrenal tumors may indeed produce small amounts of glucocorticoids that have metabolic and cardiovascular implications [10].

Concurrently, clinical consequences of both PACS/ACS involve effects especially on the cardiovascular system, metabolism, and bone architecture [32].

In ACS, the pathogenesis of cardiometabolic events appears to be the consequence of hemodynamic changes, vascular inflammatory pathways initiated and maintained by hypercortisolic status, pancreatic β-cell dysfunction, insulin resistance, and visceral obesity [28,32,62].

There is evidence that cortisol excess in these specific endocrine disorders might be associated with comorbidities such as glucose intolerance or diabetes, hypertension, osteoporosis, obesity, and dyslipidemia [2].

Furthermore, hypertension, coronary heart disease, stroke, increased left ventricular hypertrophy, subclinical atherosclerosis, arterial stiffness, and fatal or nonfatal myocardial infarction are the cardiovascular morbidities that are cited in association with cortisol excess [60,62,63].

Some authors postulated that in comparison with patients with non-functional AI, those with ACS had a prevalence of cardiovascular events that was more than three times higher. Similarly, it was discovered that ACS was linked to greater rates of diabetes and hypertension [64].

Increased cardiovascular morbidity (43% vs. 8.8%) and mortality (22.6% vs. 2.5%) in ACS patients compared to those with nonfunctioning AI have been described, with cardiovascular events and pulmonary infections as the two leading causes of death [6,64].

It is widely known that diabetes mellitus and glucose intolerance are frequent symptoms of overt Cushing’s syndrome. Furthermore, improved glycemic control (and, in some cases, diabetes reversal) following surgery has been reported to occur in 10 to 69 percent of patients with ACS who have impaired glucose tolerance or diabetes [65,66].

Patients with AI have likely diminished bone mass due to PACS/ACS, as osteoporosis is a well-known side effect of endogenous and exogenous glucocorticoid excess. In terms of bone fracture, a meta-analysis revealed that individuals with ACS had a 63.6 percent prevalence of vertebral fracture [67]. On the contrary, other authors concluded that the slight glucocorticoid excess associated with AI does not increase the risk of osteoporosis [68]. Additional factors, including decreased androgens and sarcopenia, appear to increase the risk of fractures, but further studies are needed to elucidate the exact correlations [28].

There have been reports of metabolic improvement following adrenalectomy, including weight loss, reduced blood pressure, improved glucose tolerance, and decreased cholesterol levels [69]. Moreover, adrenalectomy improved cardiovascular prognosis and death in ACS patients, according to a study [64].

A personalized strategy must take into account the level of hypercortisolism within the ACS definition, the severity of the clinical characteristics described previously, and the risks of surgery, especially in an elderly patient [6].

Furthermore, in the clinical picture of ACS, thromboembolic events and infectious diseases were also described, through direct and indirect effects of cortisol. Moreover, a considerable clinical impact appears to be played by the altered circadian cortisol rhythm and co-secretion of aldosterone and steroid precursors cited in ACS [28].

### 3.6. Hormone Production and Chromogranin a in AIs

Adrenal tumors can secrete hormones and induce clinical syndromes. Although most AI are clinically quiet, some could be functional in terms of secretion of different biochemical substances. The non-specific marker of neuroendocrine tumors is chromogranin A (CgA). Chromaffin cells in the adrenal medulla produce CgA. Studies on the plasma levels of CgA in adrenocortical adenomas have reported inconsistent results. Because it is not expressed at the immuno-histochemical level and has not been related to noticeably increased plasma levels associated with adrenocortical tumors, CgA does not appear to be involved in cortical carcinogenesis, according to multiple studies. Other researchers, however, did not discover any connection between adrenocortical adenomas and high serum CgA concentrations, even when the tumors were stained with the protein using immunohistochemistry. On the contrary, some studies suggested that cortisol-secreting adenomas had higher blood CgA values. Some authors postulated that high CgA levels are not a reliable predictor of AI’s risk for malignancy [38,70,71,72,73,74].

In addition, in the study of steroid metabolome analysis, the patients with primary aldosteronism have proven to have a glucocorticoid excess, using mass spectrometry to examine the 24 h urinary output of steroid metabolites. The cosecretion was demonstrated biochemically by the rise in various glucocorticoid precursor levels in urine samples and clinically by the onset of adrenal insufficiency following surgery [75]. Particularly, the cosecretion of adrenal androgens and glucocorticoid metabolites was cited as developing into a sensitive diagnostic tool for adrenocortical carcinoma [6].

## 4. Discussion

The evaluation and management of adrenal incidentalomas have entered a new phase with the adoption of updated European recommendations, which emphasize a risk-stratified, individualized approach. The diagnostic paradigm now integrates high-resolution imaging with structured biochemical assessment to determine both hormonal activity and malignancy potential, avoiding unnecessary interventions while ensuring timely treatment of clinically significant lesions.

A major conceptual change is the refined classification of cortisol excess. The term mild autonomous cortisol secretion (MACS) now describes patients without overt Cushingoid features but with abnormal cortisol suppression on the 1 mg overnight dexamethasone test. Cortisol ≤ 50 nmol/L (≤1.8 µg/dL) reliably excludes autonomy, while values > 138 nmol/L (>5 µg/dL) indicate clear cortisol secretion. Intermediate results define possible ACS and require correlation with clinical and metabolic findings. All patients with MACS should be evaluated for hypertension, diabetes, dyslipidemia, obesity, and osteoporosis, as even low-grade cortisol excess is linked to adverse cardiometabolic outcomes.

Imaging continues to be the primary pillar for morphological characterization of incidental lesions, with unenhanced CT attenuation thresholds guiding initial risk stratification; while traditionally ≤ 10 HU on non-contrast CT indicates lipid-rich benignity, recent paradigm work argues for nuanced, size-adapted thresholds [48,76]. Lesions that fall outside clear benign criteria—for example, those with HU > 10, heterogeneous internal architecture, or atypical enhancement/washout—warrant multidisciplinary appraisal and an individualized strategy of surveillance (commonly 6–12 months) versus definitive resection if interval growth or worrisome features appear. This patient-centered, team-based approach is emphasized in contemporary management frameworks for adrenal incidental lesions to avoid unnecessary procedures while preserving oncologic safety.

Functional and advanced anatomic techniques provide complementary information: chemical-shift MRI and dynamic washout CT reliably discriminate many benign adrenal entities, and FDG-PET supplies metabolic data that can increase diagnostic confidence for malignancy—yet FDG-PET’s likelihood ratios fall short of being definitive on their own, so metabolic imaging should be interpreted together with CT/MR morphologic features (and lesion size) rather than used in isolation. Emerging targeted PET ligands (steroid-directed tracers) and other molecular probes are under active evaluation to improve specificity for adrenocortical malignancy [48,54,76,77].

Concurrently, quantitative and molecular adjuncts are moving from research into clinical translation: CT/MR radiomics and machine-learning classifiers can extract subvisual texture and shape signatures that improve discrimination between adenoma, metastasis and primary malignancy, while urine steroid metabolomics has demonstrated high sensitivity for detecting adrenocortical carcinoma in validation studies—both approaches show promise for risk stratification but are not yet universally adopted as routine diagnostics. Integration of these tools into multidisciplinary algorithms promises to reduce unnecessary imaging/biopsy and to target interventions to patients at real risk [54,77]. Surgical management has also evolved. Adrenalectomy is indicated for hormonally active tumors, masses with radiologic suspicion of malignancy, or lesions demonstrating significant growth on surveillance.

In MACS, surgery should be considered on an individualized basis for patients with cortisol-related comorbidities, particularly younger individuals or those with progressive metabolic disease. The updated recommendations also recognize the role of partial adrenalectomy in selected cases—particularly in bilateral disease, hereditary syndromes, or when preservation of adrenal function is desirable—provided the procedure is performed in high-volume centers with endocrine expertise [3]. Conversely, surgery is discouraged for non-functional, benign-appearing unilateral adenomas, reflecting an effort to minimize overtreatment.

Management of bilateral adrenal incidentalomas requires careful hormonal assessment to exclude pheochromocytoma, hyperaldosteronism, and cortisol excess, as well as consideration of genetic testing in appropriate patients. In bilateral cortisol-producing disease, staged or partial adrenalectomy may be appropriate to avoid permanent adrenal insufficiency. Furthermore, all patients with large bilateral lesions or metastases should be screened for adrenal insufficiency, as recommended by the current guideline.

Follow-up strategies are now more clearly defined. Lesions meeting unequivocally benign criteria (≤10 HU, homogeneous) do not require repeat imaging. Indeterminate lesions not resected should be re-evaluated with imaging at 6–12 months, with surgery indicated if growth exceeds 20% (and at least 5 mm in diameter). Routine repeat hormonal testing is no longer advised if the initial biochemical evaluation is completely normal, unless new symptoms arise or comorbidities worsen. Special populations—children, young adults, and pregnant women—warrant urgent assessment, with MRI preferred over CT to reduce radiation exposure.

Despite these advances, several controversies persist. The optimal surgical threshold for MACS, the long-term cardiovascular benefit of adrenalectomy, and standardized use of novel imaging or molecular diagnostics remain unresolved. Further multicenter prospective studies are needed to define which patients with mild cortisol excess truly benefit from surgical intervention, to determine cost-effective surveillance protocols, and to validate new diagnostic technologies such as artificial intelligence-based imaging analysis and integrated molecular risk scoring [54,77].

## 5. Conclusions

Over the past few years, the understanding of adrenal incidentalomas has advanced considerably, guided by updated European recommendations that emphasize a structured yet individualized approach. Although most adrenal incidentalomas are non-functioning adrenocortical adenomas, every patient requires systematic hormonal evaluation, with specific exclusion of pheochromocytoma and cortisol excess in all cases, and screening for primary aldosteronism in hypertensive or hypokalemic individuals. Mild autonomous cortisol secretion is now recognized as the most frequent functional abnormality, carrying significant cardiometabolic implications despite the absence of overt Cushingoid features.

Imaging criteria have been refined. Indeterminate lesions require multidisciplinary review and, if not resected, interval imaging at 6–12 months, with surgery considered in the case of significant growth or suspicious features. The gold standard therapy for unilateral, hormonally active tumors or adrenocortical carcinoma remains adrenalectomy, while partial adrenalectomy is an emerging option in bilateral disease or hereditary syndromes when adrenal preservation is critical.

Pediatric and young adult patients require urgent evaluation due to the higher prevalence of malignancy, with MRI preferred to limit radiation exposure. For all patients, optimal care requires close collaboration among endocrinologists, radiologists, surgeons, and pathologists. This multidisciplinary framework supports truly personalized management, integrating radiological morphology, hormonal activity, comorbidity profile, and patient preference to avoid both overtreatment of benign lesions and undertreatment of clinically significant disease.

Despite substantial progress, major uncertainties remain regarding the long-term outcomes of surgical intervention for mild cortisol excess, optimal surveillance strategies, and the role of emerging diagnostic tools such as urine steroid metabolomics, radiomics, and artificial intelligence-based imaging analysis. Prospective multicenter studies are essential to refine surgical thresholds, validate new biomarkers, and establish cost-effective follow-up protocols.

Overall, the modern management of adrenal incidentalomas reflects a transition toward evidence-based precision medicine, aiming to balance oncological safety with avoidance of unnecessary intervention and to deliver tailored therapy for every patient.

## Figures and Tables

**Table 1 biomedicines-13-02298-t001:** Functional and non-functional adrenal masses.

Adrenal Masses Associated with Hormonal Activity	Adrenal Masses Without Hormonal Secretion
Adrenal Adenoma–cortisol/aldosterone secretion	Lymphoma
Pheochromocytoma	Metastases
Primary bilateral macronodular adrenal hyperplasia	Myelolipoma
Nodular variant of Cushing’s disease	Neuroblastoma
Congenital adrenal hyperplasia	Hemangioma
Adrenal carcinoma	Cyst
Adrenal masses associated with hormonal activity	Hemorrhage
	Granuloma
	Amyloidosis
	Ganglioneuroma
	Infiltrative disease

## Data Availability

This article is a review and does not report or generate any new research data. Therefore, no new datasets were created or analyzed in this study. All data referenced in this review are available from the sources cited throughout the article.

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
