# Peer review of "Adrenal Incidentaloma: From Silent Diagnosis to Clinical Challenge"

_biomedicines, 2025, doi:10.3390/biomedicines13092298_

Round 1

Reviewer 1 Report

Comments and Suggestions for Authors

I read with great interest this review on Adrenal incidentaloma.

Below my suggestions to improve the manuscript:

  • Check the abstract; if unstructured, please remove "Objective:" on line 18
  • I believe this introduction is too brief and should be enhanced, as well as the discussion section e.g., a paragraph on when partial adrenalectomy can be considered is needed. I suggest relying on DOI: 10.3390/jcm11051263.
  • 3.6. Particular diagnostic markers in AI , this title can be reformulated.
  • A PRISMA-style diagram needs a legend and to be numbered.

Author Response

Dear Reviewer,

Thank you very much for your careful reading and invaluable suggestions. I truly appreciate the time you invested in improving this manuscript — your comments were precise, constructive, and much appreciated.

Please find below the changes I have made in response to your points:

• Abstract (line 18): I have removed the heading “Objective:” as requested.
• Introduction: I have expanded and adapted the Introduction using your suggestion and the recommended reference to better frame the scope and rationale of the review.
• Discussion / Partial adrenalectomy: The Discussion has been strengthened and now includes a focused paragraph on when partial adrenalectomy may be considered, including key selection criteria and relevant caveats.
• Section title 3.6: I have reformulated the title “3.6. Particular diagnostic markers in AI” to a clearer heading: “3.6. Hormone Production and Chromogranin A in AIs.
• PRISMA-style diagram: I removed the PRISMA-style flowchart. This manuscript began as a chapter for a book and evolved into a narrative literature review; on reflection, a PRISMA diagram is not congruent with the narrative scope and methodology of the current paper. If you prefer that a flow diagram be retained, I will be happy to reinstate it with an appropriate legend and numbering as you suggested.

Once again, thank you from the bottom of my heart for your thoughtful critique and for the time you devoted to these suggestions. Your input has materially improved the clarity and clinical relevance of the manuscript.

With sincere appreciation,
Dr. Dana-Mihaela Tilici

Reviewer 2 Report

Comments and Suggestions for Authors

Second sentence unclear.

Add years of publication of referenced articles to reveal the historical dynamics of AI diagnosis in the third and fourth sentences.

Line 87 – add reference.

Line 90 – metastasis does not require surgical treatment.

Lines 90 – 93 – carefully reconsider this statement with other references, move to the pathogenesis part.

The pathogenesis paragraph, particularly regarding genetic predisposition, needs revision, as it appears to include only arbitrary facts.

The paragraph 3.3 Diagnostic features finally reveals that the authors have completely omitted the latest guideline from ESE/ENSAT on adrenal incidentaloma (2023) and instead refer to the previous guideline from 2016. 

The references list includes the last reference from 2021, although the authors stated that they reviewed literature up to 01 August 2025. 

This manuscript is obsolete and is not suitable for publication in 2025. 

Author Response

Dear Reviewer,

We would like to sincerely thank you for your thorough and insightful review of our manuscript. We greatly appreciate the time and effort you invested in providing such detailed feedback, which has been invaluable in improving the quality of our work.

We humbly apologize for the shortcomings you highlighted. Following your comments, we have carefully revised the manuscript to address all the points you raised. We clarified the second sentence and added the years of publication for the referenced articles in the third and fourth sentences to better reflect the historical dynamics of AI diagnosis. References have been added and updated. Most importantly, we have incorporated the latest ESE/ENSAT 2023 guideline on adrenal incidentaloma, replacing outdated references from 2016, and ensured the references list is now up-to-date through August 2025.

We would also like to provide some context regarding the evolution of this manuscript. It initially began as a chapter in a book, but during the process of reviewing literature and discussing with our co-author, we realized that the content could be more effectively presented as a narrative review article. We believe that the revised version now reflects a cohesive and updated narrative aligned with current guidelines.

We respectfully request your reconsideration of our manuscript for publication. We have carefully addressed all your concerns and sincerely hope that the revisions meet the high standards you expect. We are deeply grateful for your guidance and apologize once again for any oversights in the initial submission.

Thank you very much for your time, consideration, and understanding.

With the utmost respect and gratitude,

Dr. Dana-Mihaela Tilici

Reviewer 3 Report

Comments and Suggestions for Authors

The paper entitled "Adrenal Incidentaloma: From Silent Diagnosis to Clinical Challenge" is structured in detail and includes a large number of published studies related to adrenal incidentalomas. Most adrenal incidentalomas are nonfunctional and benign, but a subset of patients has functional and/or malignant tumors. 

The subject discussed is of great relevance in terms of impact on medical practice.

These incidentally detected lesions, called adrenal incidentalomas, have become a common clinical problem and need to be investigated for evidence of hormonal hypersecretion and/or malignancy.  The cited references contains recent publications (within the last 5 years) and relevant for the subject.

The conclusions are consistent with the evidence presented, very useful in clinical practice.

The paper is adequately presented but a revision of the English language is necessary as well as a correction related to the bibliography (37 and 38 being incomplete).

Author Response

Dear Reviewer,

We feel truly honored and exceptionally grateful that our manuscript had the privilege of reaching you and that you took the time to read it so thoroughly. It is with great humility that we receive your thoughtful and highly encouraging comments. As authors, we cannot overstate how meaningful it is for us that someone of your expertise has engaged so carefully with our work. Knowing that our manuscript was in your hands and considered for review fills us with immense appreciation and motivation.

Your recognition of the detailed structure, comprehensive coverage of the literature, and clinical relevance of our work is profoundly gratifying. We are sincerely thankful not only for your careful reading but also for the insightful and constructive feedback that you generously provided. Your thoughtful comments have given us an invaluable opportunity to reflect, refine, and improve our manuscript, and we are truly grateful for this rare and significant chance to have our work examined by someone of your esteemed experience.

In response to your suggestions, we have undertaken careful revisions of the manuscript:

1. The English language has been meticulously reviewed and enhanced throughout to improve clarity, readability, and overall presentation.

2. References 37 and 38 have been corrected, updated, and aligned with the latest available evidence.

3. Most importantly, we have updated the manuscript to fully reflect the most recent ESE/ENSAT 2023 guidelines on adrenal incidentaloma, ensuring that our recommendations and discussions are consistent with the current best practices.

We remain deeply grateful for your kind words, constructive guidance, and the time you dedicated to improving our work. Your encouragement is both humbling and inspiring, and it motivates us to ensure that our manuscript reaches the highest standards of accuracy, clinical utility, and scholarly rigor.

Please accept our heartfelt thanks for your generous support, time, and expertise. We are profoundly appreciative that our work was fortunate enough to reach you, and we hope that the revised manuscript meets your expectations and reflects the valuable insights you shared. We sincerely hope that you may find the revised manuscript suitable for acceptance in its current form.

With our warmest regards, deepest respect, and sincere gratitude,

Dr. Dana-Mihaela Tilici

On behalft of all authors

Round 2

Reviewer 1 Report

Comments and Suggestions for Authors

Authors have addressed my main concerns. I endorse acceptance,

Author Response

Dear Reviewer,

We sincerely thank you for your continued support and for taking the time to review our resubmitted manuscript.

Your feedback has been invaluable throughout this process, and we deeply appreciate your encouragement.

With gratitude,

Dr. Dana Mihaela Tilici

Reviewer 2 Report

Comments and Suggestions for Authors

Your revised manuscript indicates that you have not yet carefully engaged with the recent international guidelines on adrenal incidentaloma. A review article must be based on the latest evidence and demonstrate the author’s ability to assess and synthesize available knowledge critically. At present, your manuscript does not meet these requirements.

Author Response

Dear Reviewer,

We sincerely thank you for your meticulous and thoughtful review of our manuscript. Your insightful comments have been invaluable in helping us critically reassess our work, and we are deeply grateful for the time and expertise you have dedicated to improving it.

We wholeheartedly acknowledge your concern regarding the integration of the most recent international guidelines on adrenal incidentalomas. Following your observations, we have carefully and comprehensively revised the manuscript to fully align with the latest 2023–2025 recommendations from the European Society of Endocrinology, ENSAT, and the European Society of Cardiology, incorporating up-to-date evidence on epidemiology, imaging, biochemical evaluation, surgical management, and long-term follow-up strategies. All sections have been meticulously updated, and we have ensured that the discussion synthesizes current knowledge critically while maintaining a coherent narrative throughout.

We sincerely hope that these revisions adequately address your concerns and demonstrate our commitment to producing a review of the highest quality, fully grounded in contemporary evidence and international consensus. Your guidance has been instrumental in shaping this improved version, and we are profoundly grateful for your constructive input.

It would be a tremendous honor for us if you could consider granting the opportunity to publish this work.

We are more than willing to incorporate any additional suggestions you may have to ensure the manuscript meets the highest scholarly standards. Your approval would mean a great deal to us, and we humbly hope our revisions reflect the thorough engagement with the guidelines and critical synthesis you emphasized.

Thank you once again for your patience, guidance, and consideration.

We deeply appreciate your time and expertise, and we are sincerely hopeful for your favorable assessment of our revised manuscript.

With the utmost respect and gratitude,

Dr. Dana-Mihaela Tilici

On behalf af all authors